# Feasibility, acceptability, and effectiveness of group antenatal care on the continuum of care and perinatal outcomes in Sub-Saharan Africa: A systematic review and meta-analysis protocol

Meresa Berwo Mengesha[ID][1]*, Tesfaye Temesgen Chekole[2], Hiluf Ebuy Abraha[3,4], Etsay Weldekidan Tsegay[5], Abadi Hailay Atsbaha[6], Mihretab Gebreslassie[7,8], Zenawi Hagos Gufue[ID][6]

1 Department of Midwifery, College of Medicine and Health Sciences, Adigrat University, Adigrat, Ethiopia, 2 Department of Midwifery, Unit of Maternity and Reproductive Health, College of Medicine and Health Sciences, Dilla University, Dilla, Ethiopia, 3 Clinical Governance and Quality Improvement Unit, Ayder Comprehensive Specialized Hospital, College of Health Science, Mekelle University, Mekelle, Ethiopia, 4 Department of Epidemiology and Biostatistics, Arnold School of Public Health, University of South Carolina, Columbia, United States of America, 5 Department of Modern and Traditional Medicine Research, Tigray Health Research Institute, Mekelle, Ethiopia, 6 Department of Public Health, College of Medicine and Health Sciences, Adigrat University, Adigrat, Ethiopia, 7 Center for Epidemiology and Community Medicine, Stockholm, Sweden, 8 Department of Global Public Health, Karolinska Institute, Stockholm, Sweden

* meresaberu@gmail.com

## Abstract

### Introduction

Women with adequate antenatal care (ANC) experience more reduction in adverse maternal and perinatal outcomes compared to those with insufficient care. However, the conventional individualized ANC models have not significantly improved perinatal outcomes. Comprehensive, woman-centered group ANC (G-ANC) interventions, integrating medical care with education, demonstrate positive effects on maternal and newborn health. While promising evidence exists in sub-Saharan Africa, the feasibility, acceptability, and effectiveness of G-ANC in resource-limited settings require further investigation. The variability in current studies further indicated the need for meta-analyses and systematic reviews to consolidate findings and clarify the overall effectiveness of G-ANC interventions. This synthesis aims to provide comprehensive evidence supporting the implementation of group prenatal care models in low-resource settings. Ultimately, it seeks to establish robust evidence to guide policy and practice, contributing to reduced maternal and perinatal mortality in the region.

### Methods and analysis

This systematic review and meta-analysis protocol adheres to the Preferred Reporting Items for Systematic Reviews and Meta-Analysis Protocols guidelines. A comprehensive

**Data availability statement:** research data will be made publicly available when the study is completed and published

**Funding:** The author(s) received no specific funding for this work.;

**Competing interests:** The authors have declared that no competing interests exist.

literature search will be conducted across multiple electronic databases, including PubMed/MEDLINE, Web of Science, EMBASE, and CINHAL, to identify pertinent articles published from January 1, 2016, to June 30, 2024. Experimental studies (pre-post, quasi-experimental study, cluster randomized controlled trial), prospective cohort design, prospective comparative study, and qualitative and mixed method designs will be included in the review. The Abstract and full-text screening will be conducted by three reviewers using Covidence, according to the eligibility criteria set. The Joanna Briggs Institute (JBI) Critical Appraisal Tools, specifically designed for JBI Systematic Reviews, will be utilized to assess the methodological quality of the included studies. Statistical heterogeneity will be assessed using the Higgins test. Meta-analysis will be conducted using R statistical software version 4.4.2, which will implement random effects models to determine the weights. Pre-specified subgroup analysis and sensitivity analysis will be performed as needed. The study results will be reported in order, starting with primary outcomes and then secondary outcomes and important subgroup outcomes analyses.

## Ethics and dissemination

Ethical approval is not applicable as no original data will be collected. The findings of this review will be disseminated through publication and conference presentations.

## PROSPERO registration number

CRD42024565501

## Introduction

Maternal and perinatal mortality remains a significant global health issue, particularly in low- and middle-income countries (LMICs) [1]. According to the World Health Organization (WHO), approximately 287,000 women died from pregnancy-related causes in 2020, with Sub-Saharan Africa (SSA) bearing over 70% of this burden [1]. The region's high mortality rate is largely due to inadequate access to quality maternal healthcare, including antenatal care (ANC) [2]. ANC is considered a crucial strategy for addressing these challenges, offering preventive services, early complication detection, and vital health education [2]. Studies indicated that women receiving adequate and comprehensive ANC services have a 79% lower risk of adverse maternal and perinatal outcomes compared to those with insufficient care [3].

ANC uptake is a crucial indicator for evaluating progress towards improving maternal outcomes [4]. Comprehensive and women centered ANC interventions, which combine medical care with educational elements, have shown positive effects on maternal and newborn health outcomes [4]. The provision of high-quality, women-centered ANC is especially crucial in low- and middle-income countries (LMICs), where maternal and perinatal outcomes are often disproportionately poor [5–7]. In LMICs, particularly in SSA, the predominant model of individualized, one-on-one care has not significantly improved perinatal outcomes [8]. In contrast, group ANC (G-ANC) has emerged as a viable alternative service delivery model in high-income countries, linked to increased attendance, improved satisfaction, and better health outcomes for pregnant women and newborns [6,9,10]. This model of care also benefits women from marginalized populations who experience maternal and perinatal outcomes comparable to those observed in certain low- and middle-income countries [11,12].

The G-ANC is a transformative service delivery model that provides care to groups of eight to twelve pregnant women who are at similar gestational ages through cascades of scheduled meetings [13]. This model incorporates physical assessments, education, skill development, and peer support and takes a more holistic, woman-centered approach in contrast to traditional ANC [14]. The WHO recommends G-ANC as an alternative to individual ANC, based on rigorous research and its contextual guidance promoting community mobilization through facilitated participatory learning and action cycles [4].

The Global G-ANC Collaborative acknowledges that it is crucial to adapt G-ANC models to the unique local contexts and priorities of LMICs to guarantee ownership, sustainability, and expansion [14]. However, several challenges hinder the implementation of G-ANC in various settings, including recruiting and retaining participants, inadequate training and resources, a lack of focus on individual needs, financial barriers, and poor access to healthcare [15–17]. These challenges can be overcome by developing effective recruitment and retention strategies, utilizing mixed methods to assess fidelity and investigate the potential of G-ANC-facilitated community groups, and implementing cost-effective measures [18–20].

Various studies have demonstrated that, as compared to the individualized care model, G-ANC is associated with increased attendance at ANC visits, improved quality of care, higher rates of facility-based deliveries, enhanced health literacy and client satisfaction, increased uptake of family planning methods, better birth weights, and higher rates of breastfeeding initiation and duration [21–23]. Although the benefits of G-ANC have been demonstrated in developed countries, its feasibility, acceptability, and effectiveness in LMICs, particularly in SSA, have yet to be fully researched. However, individual studies have reported encouraging results regarding the model's efficacy in these areas, indicating the potential of G-ANC to enhance maternal and neonatal health outcomes in low-income settings [24,25].

Moreover, GANC is not widely implemented in SSA, and there is insufficient evidence regarding its effectiveness in reducing maternal and perinatal mortality in the region. Furthermore, the disparity in maternal mortality rates between developing and developed countries highlights the pressing need for targeted interventions. However, there is a lack of comprehensive meta-analyses and systematic reviews evaluating GANC's effectiveness across various settings. Current studies show considerable variability, emphasizing the need to consolidate findings from multiple research efforts to provide a clearer understanding of the overall effectiveness of group antenatal interventions. This synthesis will create a robust evidence base to inform policy and practice, ultimately helping to reduce maternal and perinatal mortality in the region.

Thus, performing a thorough review of these studies is crucial to evaluate the quality of the emerging evidence and to synthesize the available knowledge both quantitatively and qualitatively, assisting in the implementation of G-ANC models in sub-Saharan African countries. Specifically, the objective of this review is 1) to assess the effectiveness of G-ANC in increasing ANC retention and attendance, facility-based deliveries, attendance of postnatal care visits, and uptake of postpartum family planning and determine the impact of G-ANC on perinatal outcomes and the utilization of other maternal health services; 2) to synthesize the available evidence on the feasibility of G-ANC service delivery models in low-resource settings; 3) to evaluate its cultural sensitivity and acceptability among pregnant women, healthcare provider and community health workers; quality of care outcomes; and behavioral and social outcomes.

## Methods

### Eligibility criteria

**Population.**  We plan to include population groups primarily focusing on pregnant women, whether in their first or subsequent pregnancies, who are receiving antenatal care.

This will encompass adolescents and young women, women from rural areas, women with low socioeconomic status, women with limited education, HIV-positive women, multiparous and primiparous women, women from diverse ethnic and cultural backgrounds, and women with high-risk pregnancies who live in SSA. Additionally, we will consider healthcare providers, community health workers, and traditional birth attendants, who play essential roles in facilitating group antenatal care.

### Exposure

**Group Antenatal Care (G-ANC).** The intervention is G-ANC, which is defined as the integration of conventional antenatal assessments with group discussions and support. Typically, it involves 8–12 women who are at similar gestational ages of pregnancy and are guided by 1–2 group leaders. These leaders adopt a highly participatory and facilitative approach during the G-ANC sessions. The number of sessions can be adjusted to align with both global and local guidelines regarding the required frequency of visits. Each session is structured to include clinical care, client education, and ongoing discussions. During these meetings, healthcare providers deliver clinical care, and participants engage in self-assessments, such as monitoring their blood pressure and weight or identifying any warning signs. These group meetings count as antenatal care visits. The same group of women and facilitators attends all session's together, fostering stability, trust, and a sense of community among the participants.

### Comparator

For this systematic review and meta-analysis protocol, the reference group comprises women whose ANC was provided by individualized, standard, conventional, or traditional models of care.

### Outcomes

**Primary outcomes.** The primary outcomes will include the effect of G-ANC on attendance for four or more antenatal care visits, the utilization of institutional delivery, impacts on perinatal outcomes (such as birth weight, preterm birth, neonatal intensive care unit admission, and gestational age), attendance at postnatal care visits, uptake of any postpartum family planning methods, and the acceptability and feasibility of this model of care for pregnant women, healthcare providers, and community health workers.

**Secondary outcomes.** The effect of group ANC/PNC on quality of care outcomes, maternal health outcomes (including the adoption of healthy behaviors, enhanced health literacy and self-efficacy, and access to additional maternal health services), as well as behavioral and social outcomes, cultural sensitivity, and acceptability issues, will be considered secondary outcomes.

### Setting and language

The settings for this systematic review and meta-analysis protocol will encompass rural, urban, and peri-urban areas and various types of health facilities, including primary health centers, health posts, and community-based clinics. We will consider publications from January 1, 2016, to June 30, 2024, specifically those published in English. Furthermore, we will incorporate the reference lists of studies using both retrospective reference harvesting and prospective forward citation searching techniques.

### Study design

The systematic review and meta-analysis protocol will encompass experimental studies (pre-post, quasi-experimental studies, cluster randomized controlled trials), prospective cohort

designs, prospective comparative studies, and studies utilizing qualitative and mixed methods approaches.

### Exclusion

This systematic review and meta-analysis protocol will exclude preprints, unpublished reviews, conference proceedings, commentaries, editor's letters, and non-English publications.

### Information sources

A comprehensive literature search will be conducted across various electronic databases, including PubMed/MEDLINE, Web of Science, EMBASE, and CINAHL, to identify relevant articles. Information sources will be updated before submission to ensure that all pertinent studies are included. The reference lists of selected studies and systematic reviews with a similar focus will also be examined to guarantee the inclusion of all relevant studies.

### Search strategy

The search strategy for identifying relevant literature will be developed using a combination of Medical Subject Headings (MeSH) and keyword terms related to both the exposure and outcome of interest. In addition to the MeSH terms, this systematic review and meta-analysis protocol will incorporate the PICO framework (Population, Intervention, Comparator, Outcome). This method is particularly suitable for our research, which focuses on the effect of Group Antenatal Care (G-ANC) on maternal and perinatal outcomes in low-resource settings.

We will apply the PICO framework in our review protocol as follows: Population: This targets expectant mothers, especially those in resource-limited environments or from disadvantaged groups; Intervention: The intervention is G-ANC, defined as the integration of conventional antenatal assessments with group discussions and support. Comparator: We will compare G-ANC with those whose antenatal care was provided by individualized, standard, conventional, or traditional models of care. Outcome: Primary outcomes of interest will include feasibility, perinatal outcomes, and the utilization of maternal health care services/ continuum of maternal health care, while other reported outcome events can be considered secondary outcomes.

Specific criteria will be applied to the literature search, including restrictions on date, language, and publication location. **Table 1** displays the pilot search strategies for PubMed and MEDLINE. The search will be further refined to focus on articles published in SSA. Additionally, the search will be updated toward the end of the review after validation to ensure that the PubMed/MEDLINE strategy retrieves a high proportion of eligible studies.

### Protocol

The protocol for this systematic review and meta-analysis was reported following the Preferred Reporting Items for Systematic Reviews and Meta-Analysis Protocols (PRISMA-P) guidelines [26]. It has been registered on PROSPERO under registration number CRD42024565501.

### Patient and public involvement

Patients and/or the public were not involved in the design, conduct, report or dissemination plan of this research

**Table 1. Pilot search in the PubMed/MEDLINE database, June 30, 2024.**

| Database | Search restriction | Search strategy | # |
|---|---|---|---|
| PubMed/ MEDLINE | • From January 1, 2016, to June 30, 2024<br>• English<br>• Exclude preprits | **#1 AND**<br>("Group antenatal care"[Text Word]) OR ("who antenatal care"[Text Word]) OR ("antenatal care services"[Text Word]) OR ("prenatal care"[Text Word]) OR ("participatory group"[Text Word]) OR ("positive pregnancy experiences"[Text Word]) NOT ("traditional antenatal care"[Text Word]) NOT ("individual antenatal care"[Text Word])<br>**#2 AND**<br>("Impact"[Text Word]) OR ("effect"[Text Word]) OR ("effectiveness of care"[Text Word]) AND ("continuum of care"[Text Word]) OR ("continuity of care"[Text Word]) OR ("utilization of antenatal care"[Text Word]) OR ("antenatal care attendance"[-Text Word]) OR ("facility delivery"[Text Word]) OR ("quality of care"[Text Word]) OR ("postnatal attendance"[Text Word]) AND ("maternal self-efficacy"[Text Word]) OR ("maternal outcomes"[Text Word]) OR ("fetal outcomes"[Text Word]) OR ("perinatal outcomes"[Text Word])<br>**#3 AND**<br>Angola* OR Benin* OR Botswana* OR "Burkina Faso*" OR Burundi* OR "Cape Verde*" OR Cameroon* OR "Central African Republic*" OR Chad* OR Comoros* OR Congo* OR "Democratic Republic of Congo*" OR "Republic of Cote d'Ivoire*" OR "Equatorial Guinea*" OR Eritrea* OR Eswatini* OR Swaziland* OR Ethiopia* OR Gabon* OR Gambia* OR Ghana* OR Guinea* OR Guinea-Bissau* OR Kenya* OR Lesotho* OR Liberia* OR Madagascar* OR Malawi* OR Mali* OR Mauritania* OR Mauritius* OR Mozambique* OR Namibia* OR Niger* OR Nigeria* OR Rwanda* OR "Sao Tome and Principe*" OR Senegal* OR Seychelles* OR "Sierra Leone*" OR Somalia* OR "South Africa*" OR "South Sudan*" OR Sudan* OR Tanzania* OR Togo* OR Uganda* OR Zambia* OR Zimbabwe* OR "West Africa*" OR "southern Africa*" OR "south Africa*" OR "east Africa*" OR Africa* OR "sub-Saharan Africa*" | 111 |

## Data management

The findings retrieved from the literature search will be imported, screened, and analyzed using professional software platforms such as Microsoft Excel, R statistical software, and Covidence. Covidence will facilitate the automatic elimination of duplications, coupled with a manual verification process for identifying resemblances among studies (e.g., publication year, authorship, journal details, etc.) conducted by the authors.

## Selection process

Three reviewers (MG, ZHG, and HEA) will conduct abstract and full-text screening based on the eligibility criteria using Covidence. Any discrepancies among the three reviewers will be resolved with the help of other reviewers (MBM). A separate section will be created in Excel specifically for relevant studies identified during full-text screening, including reference lists of included studies and systematic reviews within the same domain. Rationales for exclusion will be carefully recorded throughout the full-text screening process. A final determination regarding the inclusion of studies will be made by considering the outcomes from both Covidence and the spreadsheet. The presentation of study selection results will utilize PRISMA flow diagrams (PRISMA 2020 flow diagram for new systematic reviews which included searches of databases, registers and other sources, reproduced from Page et al. [27] (**Fig 1**).

## Data collection

A standardized data collection tool will be developed as a data extraction form. Subsequently, this form will undergo a pilot test among various groups, with potential modifications being made based on feedback received. Three reviewers (EWT, AHA, and TTC) will independently extract data from the studies included in the analysis. To ensure consistency in assessment methods, the reviewers will undergo a calibration exercise. In cases of discrepancies between reviewers, the principal investigator (MBM) will be consulted for resolution.

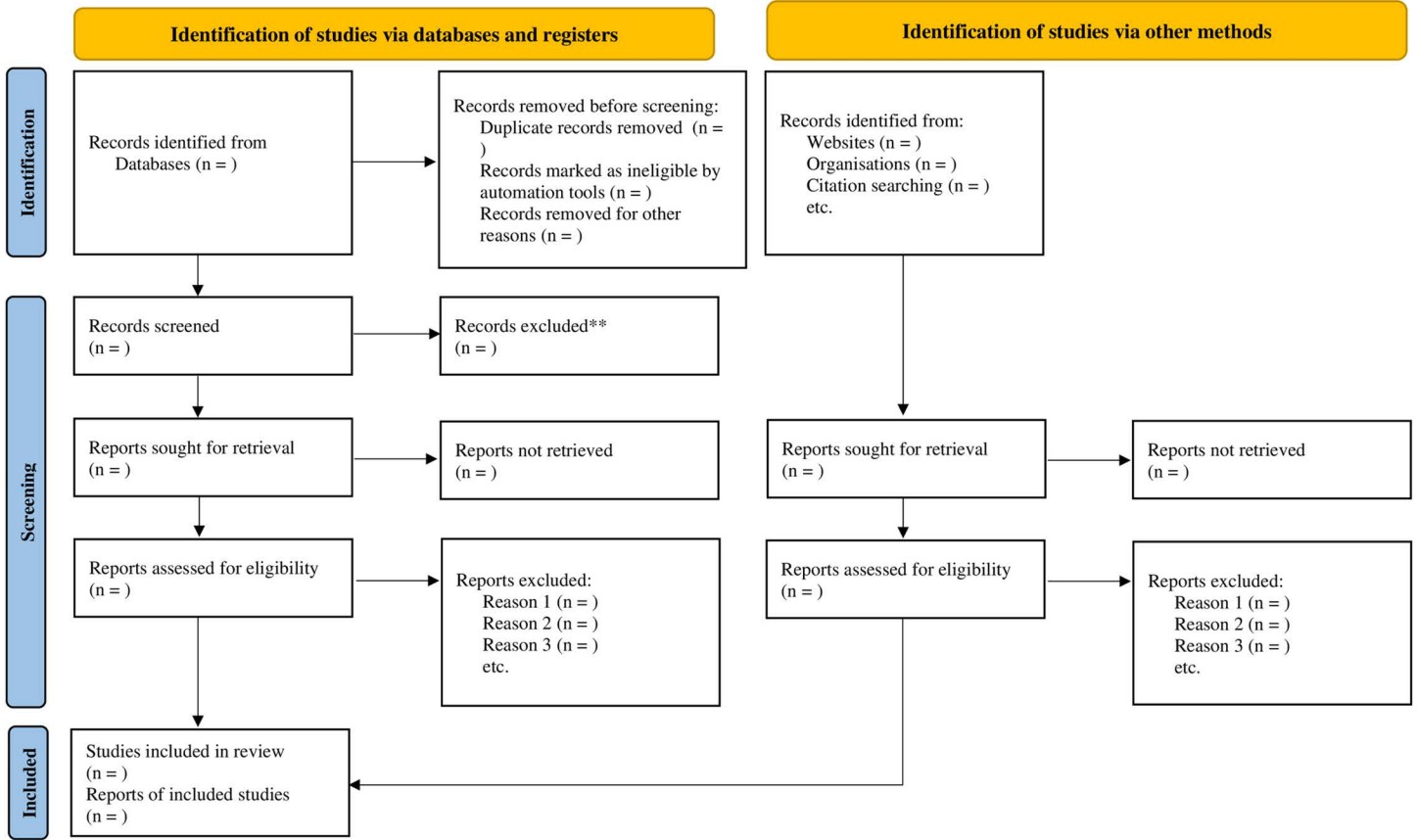

**Fig 1.** PRISMA 2020 flow diagram for new systematic reviews which included searches of databases, registers and other sources, reproduced from page et al [27].

## Data items

The following data items will be extracted from included studies: (1) Study data: title, author name, year of publication, country of study, journal, sample size, study period, study design, follow-up period (for experimental and cohort) and limitations; (2) Population: participant characteristics (number of group of women, number of women per group, how women are grouped, number of leaders in the group, who leads the group, the total number of group sessions (ANC sessions), length of the group session, etc.); (3) Intervention: G-ANC, or participatory group, involves assessing their experience, perspective, and outcomes within the group settings. (4) Comparison: the comparator used in the studies is conventional care or individualized ANC. (5) Outcomes: a composite of outcome events (feasibility, acceptability, effectiveness on perinatal outcomes, and continuum of care). (6) Effect measures: reported effect measures for the reported effect measures in the composite outcomes, either quantitatively or qualitatively synthesized, separate outcomes if available, including P-values, standard deviation, relative risk, hazard ratio, and confidence interval.

## Outcomes and prioritization

The findings from the available evidence regarding the impact of Group Antenatal Care (G-ANC) on various factors will be synthesized. These factors include attendance at four or more antenatal care visits, use of institutional delivery, perinatal outcomes (such as

birth weight, preterm birth, neonatal intensive care unit admissions, and gestational age), attendance at postnatal care visits, uptake of postpartum family planning methods, and the acceptability and feasibility of this model of care among pregnant women, healthcare providers, and community health workers. Furthermore, comprehensive data on the impact of group ANC/PNC on quality of care outcomes, maternal health outcomes (such as uptake of healthy behaviors, increased health literacy, self-efficacy, and other maternal health services), as well as behavioral and social outcomes, cultural sensitivity, and acceptability will be examined. The main outcome will focus on determining the composite of feasibility, perinatal outcomes, and the utilization of maternal healthcare services as part of the continuum of maternal healthcare. In contrast, other reported outcome events will be considered secondary outcomes.

## Risk of bias assessment

The Joanna Briggs Institute (JBI) Critical Appraisal Tools, specifically designed for JBI Systematic Reviews of both quantitative and qualitative studies, will be used to evaluate the methodological quality of the included studies [28,29]. To assess the quality of articles for cohort, non-randomized, quasi-experimental, and randomized controlled trial (RCT) studies, the JBI Critical Appraisal Tools, which include specific checklists for each study design, will be utilized. Three assessors (TTC, AHA, and EWT) will evaluate the quality of each research study, and any discrepancies will be resolved through discussion. To address limitations when using the Joanna Briggs Institute (JBI) critical appraisal tools, the following techniques will ensure quality and reliability: Comprehensive training will be provided to reviewers to effectively use the tools. Standardized guidelines or protocols will maintain consistency across reviewers for each study design. Pilot appraisals on a small set of studies will help identify potential challenges. More than three independent reviewers will evaluate studies and compare results to minimize bias. Clear documentation of decisions will enhance transparency and reproducibility.

## Data synthesis

The characteristics of pregnant women (number of groups, group size, number of women grouped, number of leaders in the group, who leads the group, the total number of group sessions (number of ANC), follow-up periods for the outcome and setups for G-ANC) will be assessed. Statistical heterogeneity will be assessed using the Higgins test, where the $I^2$ statistics will be determined and reported. If the studies included in the analysis are homogeneous, a meta-analysis will be conducted to calculate the overall effect of G-ANC on the continuum of care and perinatal outcomes compared to conventional individualized care.

This analysis will utilize R version 4.4.2 software, applying random effects to determine the weights for the meta-analysis, if necessary, after thoroughly investigating the presence and nature of heterogeneity. A qualitative synthesis will be performed if there is significant heterogeneity ($I^2 \geq 50\%$ or $P < 0.1$) or in situations where the data are incomplete or unsuitable for meta-analysis. Moreover, when appropriate, additional analytical strategies will be evaluated to strengthen the robustness and depth of the analysis, such as subgroup analysis, sensitivity analysis, or meta-regression. Furthermore, to minimize the risk of bias and ensure the reliability of the findings, studies with certain designs (e.g., high risk of bias or methodological limitations) will be excluded from the analysis.

If various types of effect measures are utilized in the original studies, such as odds ratios, risk ratios, and hazard ratios, the meta-analysis will be conducted for each type of effect

measures. The study results will be reported sequentially, commencing with primary outcomes, followed by secondary outcomes and important subgroup analysis based on design, setup, country/region context, and methods of group classification. These methods will be conducted to investigate the possible causes of variability between studies and to explore the strength of the meta-analysis.

Given that the effectiveness of G-ANC can be greatly affected by the country's context, policy, and resources, stratification will be made based on the country's context or region (Eastern Africa, Western Africa, South Africa, and Central Africa). Additionally, study results will be stratified based on the setup where the G-ANC was conducted (facility-based vs. community-based intervention). Furthermore, study results will be stratified based on study design to assess whether the study design influences the association between exposure and outcomes. The number of women in a group and how they are grouped may impact the effectiveness of G-ANC.

For qualitative synthesis, we will present summary and narrative statements and quotes from the experiences of pregnant women, group leaders, and health workers regarding the feasibility and acceptability of G-ANC.

## Meta-bias (es)

Outcome reporting biases will be assessed by comparing outcomes documented in research protocols with those reported in the actual study reports. Additionally, sensitivity analysis using STATA will evaluate the impact of selective reporting on the results of meta-analyses, if deemed necessary. Funnel plots will also be employed to investigate potential publication bias.

## Confidence in cumulative evidence

The Grading of Recommendations, Assessment, Development, and Evaluation working group methodology will be used to assess the quality of evidence for all outcomes. The assessment will consider the domains of risk of bias, consistency, directness, precision, and publication bias. Due to the mixed nature of the studies comprising this review, including both observational and experimental research, the evidence will initially be assessed as moderate. However, if a substantial effect size is present, a dose-response relationship is established, or all potential biases are minimal, the strength of the evidence may be potentially upgraded [30].

## Supporting information

**S1 File. PRISMA-P (Preferred Reporting Items for Systematic review and Meta-Analysis Protocols) 2015 checklist: recommended items to address in a systematic review protocol.** (DOCX)

## Author contributions

**Conceptualization:** Meresa Berwo Mengesha.

**Methodology:** Meresa Berwo Mengesha, Tesfaye Temesgen Chekole, Hiluf Ebuy Abraha, Etsay Weldekidan Tsegay, Abadi Hailay Atsbaha, Mihretab Gebreslassie, Zenawi Hagos Gufue.

**Writing – original draft:** Meresa Berwo Mengesha.

**Writing – review & editing:** Meresa Berwo Mengesha, Tesfaye Temesgen Chekole, Hiluf Ebuy Abraha, Etsay Weldekidan Tsegay, Abadi Hailay Atsbaha, Mihretab Gebreslassie, Zenawi Hagos Gufue.

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
