## [Decision Letter · Decision Letter 0]

5 Feb 2025

PONE-D-24-40365Feasibility, Acceptability, and Effectiveness of Group Antenatal Care on the Continuum of Care and Perinatal Outcomes in Sub-Saharan Africa: A Systematic Review and Meta-analysis ProtocolPLOS ONE

Dear Dr. Mengesha,

Thank you for submitting your manuscript to PLOS ONE. After careful consideration, we feel that it has merit but does not fully meet PLOS ONE’s publication criteria as it currently stands. Therefore, we invite you to submit a revised version of the manuscript that addresses the points raised during the review process.

We look forward to receiving your revised manuscript.

Kind regards,

Hale Teka

Academic Editor

PLOS ONE

Additional Editor Comments :

Introduction:

Please restructure this section. Start with broader topics such as maternal mortality - globally and the burden in SSA. Solicit the link in between maternal and perinatal mortality with ANC. Define both ANC and G-ANC. Then discuss the gap - GANC, then discuss what you are going to do about it - in this case systematic review and meta-analysis.

Line 59 "pregnancy and perinatal outcome" - pls revise this terminology. Perinatal outcome is a pregnancy outcome. Possible suggestion - maternal and perinatal outcome.

Line 64 - 66: "This model is also shown to benefit marginalized women, whose perinatal outcomes are comparable to those in developed countries (8, 9)." This is a confusing statement. Do you mean that the studies show that having GANC brings perinatal outcomes to the same level to those in developed countries? Please rewrite it.

Line 93 - "this review" - this is not a review yet. this is a protocol. Please review throughout the manuscript as a protocol not as a review.

Methods section:

Population - are you going to select vulnerable population in the Sub-saharan Africa or you are defining women in Sub-saharan Africa? The way you described your population is confusing.

How is feasibility your secondary outcome? "Feasibility, Acceptability, and Effectiveness of Group Antenatal Care on the Continuum of Care and Perinatal Outcomes in Sub-Saharan Africa: A Systematic Review and Meta-analysis Protocol" This is your research title. How can your primary objective be a secondary outcome? Your primary objective should be to assess feasibility, acceptability, and effectiveness.

The way you selected the study period sounds random. Why did you choose from 2016 to mid 2024? Why do you want to examine G-ANC in this time period?

Your exclusion needs revision. There wont be a case report or case series written on G-ANC - not even a chance.

Search Strategy: This section is written as if you have already conducted the systematic review and meta-analysis. Please revise it.

Reviewers' comments:

Reviewer's Responses to Questions

**Comments to the Author**

1. Does the manuscript provide a valid rationale for the proposed study, with clearly identified and justified research questions?

Reviewer #1: Yes

Reviewer #2: Yes

2. Is the protocol technically sound and planned in a manner that will lead to a meaningful outcome and allow testing the stated hypotheses?

Reviewer #1: Partly

Reviewer #2: Yes

3. Is the methodology feasible and described in sufficient detail to allow the work to be replicable?

Reviewer #1: No

Reviewer #2: No

4. Have the authors described where all data underlying the findings will be made available when the study is complete?

Reviewer #1: Yes

Reviewer #2: Yes

5. Is the manuscript presented in an intelligible fashion and written in standard English?

Reviewer #1: No

Reviewer #2: Yes

6. Review Comments to the Author

You may also provide optional suggestions and comments to authors that they might find helpful in planning their study.

Reviewer #1: I thank the Editor for letting me review the paper.

The research question is fundamental, and the protocol is suitable overall.

Abstract

You have failed to acknowledge that this is a protocol; you have structured your abstract as if you have conducted the systemic review and meta-analysis. You are advised to revise the aim statement of your abstract.

Please use ‘the planned review ‘ or ‘the review’ when addressing the systemic review and meta-analysis. Use this protocol when you are addressing the protocol.

Introduction

Please begin your paragraph with the burden of maternal mortality and perinatal mortality. Then, provide evidence that ANC is one of the core strategies for mitigating the burden. Then, in subsequent paragraphs, you argue the importance of G ANC VS ANC, geographic disparity, the paucity of meta-analysis and systemic review, and the need to conduct the study.

Again, please revise your aim statement.

Methods

Primary outcomes

Maternal morbidity and mortality are not typically considered as ‘perinatal outcomes.’ Perinatal outcome refers to the status of the fetus or newborn during the perinatal period, which is between 24-28, depending on the setup, up to seven days after birth. Usually utilized perinatal outcomes include birthweight, preterm birth, neonatal intensive care unit admission, gestational age, stillbirth, early neonatal death, and neonatal morbidities. Therefore, you should revise your statements accordingly and avoid conflating maternal and perinatal outcomes.

Please reconcile what you wrote in the ‘ Outcomes and prioritization ‘ section with primary and secondary outcomes. Furthermore, you need to be specific.

Please describe the intervention group instead of saying ‘G-ANC and/or Participatory Group.’

Inclusion / Exclusion criteria

Could you please clarify whether you will include published abstracts, conference proceedings, and reference lists of the studies that are incorporated?

Search strategies

Please improve the grammar of the paragraph; mix of tenses.

Risk of bias assessment

Please mention the techniques that will be used to mitigate the JBI critical appraisal tools. Such as training….

Thank you

Reviewer #2: Manuscript Number: PONE-D-24-40365

Manuscript Title: Feasibility, Acceptability, and Effectiveness of Group Antenatal Care on the Continuum of Care and Perinatal Outcomes in Sub-Saharan Africa: A Systematic Review and Meta-analysis Protocol

General comment

- It is well-written and very important for the area.

Specific comment

Abstract:

Line number 32-24: The justification for the importance of this systematic review and meta-analysis needs to be strengthened. The author mentions that 'pilot studies show promising evidence of its effectiveness in these areas.' To enhance this rationale, it is crucial to highlight additional factors, such as conflicting findings in existing literature and unresolved research gaps. This will help better identify existing gaps and provide stronger justification for assessing the outcomes.

Line number 47-48: In the abstract method section, it states, “Pre-specified subgroup analysis and sensitivity analysis will be conducted as necessary.” If there is heterogeneity, subgroup analysis is recommended based on scientific evidence. Therefore, why it stated as necessary.

Method :

Lines 112-117: The primary outcomes are numerous. It would be beneficial to narrow them down to a few specific, measurable outcomes that clearly define the effect measures.

Line 122-123: Setting : it will be focused on Health facility type ,rural and urban settion? Additionally, GANC expected to be effective only in rural areas or in communities where health facilities are not well implemented and accessible.

Lines 129-130: 'This review will exclude preprints, unpublished reviews, case reports, case series, commentaries, editor’s letters, and non-English publications.' One of the key principles of systematic reviews and meta-analyses is to capture both positive and negative findings, as negative results are often underreported in published literature. Since many studies, including those from pharmaceutical companies, may not publish negative findings, incorporating grey literature can strengthen the systematic review and meta-analysis by reducing publication bias.

Lines 138-139: The search strategy needs to be clearer. In addition to using MeSH terms, it would be beneficial to explain whether a framework such as PICO, SPICE, or SPIDER is used, especially since the inclusion criteria cover qualitative and mixed-method studies.

Lines 218-219: “Or if the data is incomplete, a qualitative synthesis will be conducted instead of a meta-analysis.” It would be beneficial to recommend additional analytical approaches, such as subgroup analysis, sensitivity analysis, or meta-regression, rather than solely opting for qualitative synthesis. Additionally, considering the risk of bias, studies with certain designs may be excluded to improve the robustness of the analysis.

7. PLOS authors have the option to publish the peer review history of their article (what does this mean? ). If published, this will include your full peer review and any attached files.

**Do you want your identity to be public for this peer review?** For information about this choice, including consent withdrawal, please see our Privacy Policy .

Reviewer #1: **Yes: ** Awol Yemane Legesse

Reviewer #2: No

---

## [Author Response · Author response to Decision Letter 1]

14 Feb 2025

Date: February 13, 2025.

Dear Editor-in-chief,

Subject: Submission of the corrected systematic review and meta-analysis protocol manuscript.

We wish to extend our gratitude to the editor and reviewers for their valuable and constructive comments and suggestions. Please find attached the author’s point-by-point response letter addressing the editor’s and reviewers’ comments and suggestions. Also attached are a clean protocol manuscript and the tracked changes document for the systematic review and meta-analysis protocol entitled “Feasibility, Acceptability, and Effectiveness of Group Antenatal Care on the Continuum of Care and Perinatal Outcomes in Sub-Saharan Africa: A Systematic Review and Meta-analysis Protocol” with manuscript number PONE-D-24-40365 by Mengesha MB, et al., for your consideration.

We thoroughly revised the systematic review and meta-analysis protocol in response to the editor’s and reviewers’ comments and suggestions. We confidently assert that the revised protocol meets the publication standards of your esteemed journal. We look forward to your response, appreciate your consideration, and await the outcome of your assessment.

Yours sincerely,

Meresa

Meresa Berwo Mengesha (BSc, MSc, Assistant Professor), Corresponding Author

Department of Midwifery, College of Medicine and Health Sciences,

Adigrat University, Adigrat, Ethiopia.

Email: meresaberu@gmail.com.

Response to the Academic Editor’s and reviewer’s comments

Additional Editor Comments

Introduction:

1. Please restructure this section. Start with broader topics such as maternal mortality - globally and the burden in SSA. Solicit the link in between maternal and perinatal mortality with ANC. Define both ANC and G-ANC. Then discuss the gap - GANC, then discuss what you are going to do about it - in this case systematic review and meta-analysis.

Author’s response: Dear academic editor, thank you very much for your valuable recommendation to create a catchy and structured outline for the introduction; we appreciate your comments and have revised it according to the suggested ideas.

2. Line 59 "pregnancy and perinatal outcome" - please revise this terminology. Perinatal outcome is a pregnancy outcome. Possible suggestion - maternal and perinatal outcome.

Author’s response: Thank you very much for your valuable recommendation to revise this terminology. It has been modified as per your recommendation.

3. Line 64 - 66: "This model is also shown to benefit marginalized women, whose perinatal outcomes are comparable to those in developed countries (8, 9)." This is a confusing statement. Do you mean that the studies show that having GANC brings perinatal outcomes to the same level to those in developed countries? Please rewrite it.

Author’s response: Thank you for seeking clarification on this issue. “This model is also shown to benefit marginalized women, whose perinatal outcomes are comparable to those in developed countries.” We mean that the findings indicate a comprehensive, women-centered group antenatal care intervention provided to disadvantaged women may lead to maternal and perinatal outcomes that are relatively comparable to those in developed countries. However, this does not necessarily imply the same level as in developed nations, as there are other factors to consider. It has been revised so that “relatively comparable” appears in the protocol with the tracked changes.

4. Line 93 - "this review" - this is not a review yet. this is a protocol. Please review throughout the manuscript as a protocol not as a review.

Author’s response: Thank you for the recommendation. We have revised it according to the suggested idea. It is now titled “systematic review and meta-analysis protocol."

Methods section:

5. Population - are you going to select vulnerable population in the Sub-Saharan Africa or you are defining women in Sub-Saharan Africa? The way you described your population is confusing.

Author’s response: Thank you for raising this critical issue. To clarify any confusion regarding the selection of the target population, we plan to include population groups primarily focused on pregnant women—whether in their first or subsequent pregnancies—who are receiving antenatal care. This will encompass adolescents and young women, women from rural areas, women with low socioeconomic status, women with limited education, HIV-positive women, multiparous and primiparous women, women from diverse ethnic and cultural backgrounds, and women with high-risk pregnancies. Additionally, we will consider healthcare providers, community health workers, and traditional birth attendants, who play essential roles in facilitating group antenatal care. This information is added to the population portions of the subheading under the methods section. Please review the manuscript with track changes.

6. How is feasibility your secondary outcome? "Feasibility, Acceptability, and Effectiveness of Group Antenatal Care on the Continuum of Care and Perinatal Outcomes in Sub-Saharan Africa: A Systematic Review and Meta-Analysis Protocol" This is your research title. How can your primary objective be a secondary outcome? Your primary objective should be to assess feasibility, acceptability, and effectiveness.

Author’s response: Thank you for raising this concern. We have revised it, and the following assertions have been added to the protocol with the track changes: “The primary outcomes will be the effect of G-ANC on attendance at four or more antenatal care visits, utilization of institutional delivery, impact on perinatal outcomes (birth weight, preterm birth, neonatal intensive care unit admission, gestational age), attendance at postnatal care visits, uptake of any postpartum family planning methods, and the acceptability and feasibility of this model of care by pregnant women, healthcare providers, and community health workers."

Secondary outcomes

Effect of group ANC/PNC on the Quality of care outcomes, maternal health outcomes (uptake of healthy behaviors, increased health literacy and self-efficacy and other maternal health services), Behavioral and social outcomes, cultural sensitivity and acceptability issues will be considered as secondary outcomes.”

7. The way you selected the study period sounds random. Why did you choose from 2016 to mid-2024? Why do you want to examine G-ANC in this time period?

Author’s response: We appreciate your thoughtful insight regarding the time restriction of this review. Why was the study period limited to certain years (in this case, from January 1, 2016, to June 30, 2024)? Certainly, it is confined to these years for the following reasons.

1. Since the initiation of research in this area (in 2016, the Global Group Antenatal Care Collaborative was established as a platform for group antenatal care researchers working in low- and middle-income countries to share experiences and shape future studies to enhance the development of a solid global evidence base reflecting implementation and outcomes specific to low- and middle-income countries), most studies in this specific field within LMICs have been conducted after 2016. You can read this article (Building a Global Evidence Base to Guide Policy and Implementation for Group Antenatal Care in Low‐ and Middle‐Income Countries: Key Principles and Research Framework Recommendations from the Global Group Antenatal Care Collaborative) (escholarship.org).

2. Around this time, after 2016, the World Health Organization (WHO) and other global health organizations began to emphasize the importance of person-centered care in maternal health, marking a shift toward this approach. Group ANC, which promotes peer support and active participation, aligns well with this model, prompting further research into its feasibility and effectiveness.

3. After 2015, the Sustainable Development Goals (SDGs) were launched, with SDG 3 specifically targeting maternal and child health. This led to increased funding and support for maternal health innovations, including group ANC, from global health organizations, governments, and NGOs. We aimed to see the effectiveness of this model in this case.

4. Furthermore, the release of updated WHO guidelines on antenatal care in 2016, which recommended a minimum of eight ANC contacts and emphasized the importance of quality care, provided a strong impetus for exploring innovative models like group ANC. Overall, the increase in studies on group ANC in LMICs after 2016 restricted our ability to assess the model's feasibility, acceptability, and effectiveness in low-resource settings.

8. Your exclusion needs revision. There won’t be a case report or case series written on G-ANC - not even a chance.

Author’s response: Thank you for highlighting this intriguing point. While case reports and case series are not typically used in GANC research, they could be valuable in documenting unique implementations and challenges in low-resource settings. However, to date, most existing studies have utilized larger-scale designs, such as randomized controlled trials (RCTs), mixed-methods studies, and quasi-experimental designs, to evaluate the effectiveness and feasibility of GANC. We considered including these less common designs if they offered meaningful insights into the implementation and challenges of this care model. However, after a thorough search and screening process during the review, we found no such studies. As a result, we have removed the statement regarding “case reports and case series” and updated the protocol accordingly.

9. Search Strategy: This section is written as if you have already conducted the systematic review and meta-analysis. Please revise it.

Author’s response: Thank you for your request to revise this section. We have made the necessary revisions, and we plan to address and update it at the end of the review process.

Responses to Reviewer 1:

I thank the Editor for letting me review the paper. The research question is fundamental, and the protocol is suitable overall.

Author’s response: Dear reviewer, we are grateful for your time and significant contributions to this protocol. We learned a great deal from your scientific arguments.

Abstract

1. You have failed to acknowledge that this is a protocol; you have structured your abstract as if you have conducted the systemic review and meta-analysis. You are advised to revise the aim statement of your abstract.

Author’s response: Thank you for your critical observation and for informing us about this. In our context, a planned systematic review and meta-analysis refers to the systematic review and meta-analysis protocol. We have revised it according to your recommendation. Please take a look at the revised protocol with track changes.

2. Please use ‘the planned review ‘or ‘the review’ when addressing the systemic review and meta-analysis. Use this protocol when you are addressing the protocol.

Author’s response: We have amended it according to the suggestion given for improvement.

Introduction

1. Please begin your paragraph with the burden of maternal mortality and perinatal mortality. Then, provide evidence that ANC is one of the core strategies for mitigating the burden. Then, in subsequent paragraphs, you argue the importance of G ANC VS ANC, geographic disparity, the paucity of meta-analysis and systemic review, and the need to conduct the study.

Author’s response: Thank you very much for your valuable recommendation to create a catchy and structured outline of the introduction section. We appreciate your feedback and have revised the document accordingly. Please take a look at the revised protocol with the tracked changes.

2. Again, please revise your aim statement.

Author’s response: we accept the comments and we made revisions as per the suggested idea. Kindly have a look at the revised protocol with the track changes.

Methods

Primary outcomes

1. Maternal morbidity and mortality are not typically considered as ‘perinatal outcomes.’ Perinatal outcome refers to the status of the fetus or newborn during the perinatal period, which is between 24-28, depending on the setup, up to seven days after birth. Usually utilized perinatal outcomes include birthweight, preterm birth, neonatal intensive care unit admission, gestational age, stillbirth, early neonatal death, and neonatal morbidities. Therefore, you should revise your statements accordingly and avoid conflating maternal and perinatal outcomes.

Author’s response: We appreciate your critical observation and admit that it was an editorial error. It is revised in the manuscript with the track changes according to the suggested idea.

2. Please reconcile what you wrote in the ‘Outcomes and prioritization ‘section with primary and secondary outcomes. Furthermore, you need to be specific.

Author’s response: We have checked it and revised it accordingly. The outcome and prioritization align with the above-specified primary and secondary outcomes. Please kindly have a look at the revised protocol with the track changes.

3. Please describe the intervention group instead of saying ‘G-ANC and/or Participatory Group.’

Author’s response: Thank you for reminding us to include the intervention description (G-ANC). The statement provided below has also been added to the protocol with track changes.

The intervention is G-ANC, which is defined as the integration of conventional antenatal assessments with group discussions and support. Typically involves 8 to 12 women who are at similar gestational ages of pregnancy, guided by 1 to 2 group leaders. These leaders adopt a highly participatory and facilitative approach during the G-ANC sessions. The number of sessions can be adjusted to align with both global and local guidelines regarding the required frequency of visits. Each session is structured to include clinical care, client education, and ongoing discussions. During these meetings, healthcare providers deliver clinical care, and participants engage in self-assessments, such as monitoring their blood pressure and weight or identifying any warning signs. These group meetings count as antenatal care visits. The same group of women and facilitators attends all session’s together, fostering stability, trust, and a sense of community among the participants.

Inclusion / Exclusion criteria

1. Could you please clarify whether you will include published abstracts, conference proceedings, and reference lists of the studies that are incorporated?

Author’s response: We will include published articles written in English. Additionally, we will incorporate the reference lists of studies by applying both retrospective reference harvesting and prospective forward citation searching techniques. However, we will exclude preprints that have not been extensively reviewed, unpublished reviews, conference proceedings, commentaries, editor’s letters, and non-English publications.

Search strategies

1. Please improve the grammar of the paragraph; mix of tenses.

Author’s response: Thank you for your request to revise this section. We have made the necessary revisions, as we intend to address them and update them after the review process.

Risk of bias assessment

1. Please mention the techniques that will be used to mitigate the JBI critical appraisal tools. Such as training….

Author’s response: Thank you for your recommendation to add strategies for mitigating the limitations of the JBI critical appraisal tools in the systematic review and meta-analysis. Following your suggestion, the following statements were added to the protocol with track changes: “To address the limitations of using the Joanna Briggs Institute (JBI) critical appraisal tools, the following techniques will be employed to ensure the quality and reliability of the appraisal process. Comprehensive training will be provided for reviewers on effectively using the JBI critical appraisal tools. Additionally, standardized and clear guidelines will be utilized to ensure consistency across reviewers for each study design. Moreover, we will conduct pilot appraisals on a small subset of studies to identify potential challenges or ambiguities in the tool's application. Furthermore, we will recruit three independent reviewers to appraise studies and compare results to minimize bias. F

---

## [Editor Report · Decision Letter 1]

19 Feb 2025

PONE-D-24-40365R1Feasibility, Acceptability, and Effectiveness of Group Antenatal Care on the Continuum of Care and Perinatal Outcomes in Sub-Saharan Africa: A Systematic Review and Meta-analysis ProtocolPLOS ONE

Dear Dr. Mengesha,

Thank you for submitting your manuscript to PLOS ONE. After careful consideration, we feel that it has merit but does not fully meet PLOS ONE’s publication criteria as it currently stands. Therefore, we invite you to submit a revised version of the manuscript that addresses the points raised during the review process.

Line 64 - 66: "This model is also shown to benefit marginalized women, whose perinatal outcomes are comparable to those in developed countries" 

While your descriptive response was good, the actual revision you made in the manuscript is still confusing. Please refer to the original review and revise this statement accordingly. 

We look forward to receiving your revised manuscript.

Kind regards,

Hale Teka

Academic Editor

PLOS ONE

Journal Requirements:

Additional Editor Comments:

Line 64 - 66: "This model is also shown to benefit marginalized women, whose perinatal outcomes are comparable to those in developed countries"

While your descriptive response was good, the actual revision you made is still confusing. Please revise this statement.

---

## [Author Response · Author response to Decision Letter 2]

19 Feb 2025

Date: February 19, 2025.

Dear Editor-in-chief,

Subject: Submission of the corrected systematic review and meta-analysis protocol manuscript.

We wish to extend our gratitude to the editor and reviewers for their valuable and constructive comments and suggestions. Please find attached the author’s point-by-point response letter addressing the editor’s and reviewers’ comments and suggestions. Also attached are a clean protocol manuscript and the tracked changes document for the systematic review and meta-analysis protocol entitled “Feasibility, Acceptability, and Effectiveness of Group Antenatal Care on the Continuum of Care and Perinatal Outcomes in Sub-Saharan Africa: A Systematic Review and Meta-analysis Protocol” with manuscript ID PONE-D-24-40365 by Mengesha MB, et al., for your consideration.

We thoroughly revised the systematic review and meta-analysis protocol in response to the editor’s and reviewers’ comments and suggestions. We confidently assert that the revised protocol meets the publication standards of your esteemed journal. We look forward to your response, appreciate your consideration, and await the outcome of your assessment.

Yours sincerely,

Meresa

Meresa Berwo Mengesha (BSc, MSc, Assistant Professor), Corresponding Author

Department of Midwifery, College of Medicine and Health Sciences,

Adigrat University, Adigrat, Ethiopia.

Email: meresaberu@gmail.com.

Response to reviewers and editor’s comments

Journal Requirements:

Author’s response: Thank you for your critical observation and for bringing this to our attention. We have cross-checked the references twice and confirmed that we have not cited any retracted papers. However, we have made revisions to address some instances of duplication and ensured that the reference is now complete and accurate.

Additional Editor Comments:

Line 64 - 66: "This model is also shown to benefit marginalized women; whose perinatal outcomes are comparable to those in developed countries". While your descriptive response was good, the actual revision you made is still confusing. Please revise this statement.

Author’s response: Dear editor, thank you for your feedback and seeking clarification on this issue. We have revised it accordingly. The statement provided below has also been added to the protocol with track changes. “Previous studies have shown that this care model, which provides comprehensive, women-centered group antenatal care to underserved and disadvantaged populations, has increased healthcare access while achieving maternal and perinatal outcomes comparable to those in developed countries. However, this comparison does not imply equivalence because other factors may influence the outcomes” Please take a look at the revised protocol with the tracked changes.

---

## [Editor Report · Decision Letter 2]

27 Feb 2025

PONE-D-24-40365R2Feasibility, Acceptability, and Effectiveness of Group Antenatal Care on the Continuum of Care and Perinatal Outcomes in Sub-Saharan Africa: A Systematic Review and Meta-analysis ProtocolPLOS ONE

Dear Dr. Mengesha,

Thank you for submitting your manuscript to PLOS ONE. After careful consideration, we feel that it has merit but does not fully meet PLOS ONE’s publication criteria as it currently stands. Therefore, we invite you to submit a revised version of the manuscript that addresses the points raised during the review process.

We look forward to receiving your revised manuscript.

Kind regards,

Hale Teka

Academic Editor

PLOS ONE

Journal Requirements:

**Additional Editor Comments:**

The authors' response in the point-by-point response and their response in the manuscript doesn’t match. I kindly ask them to revisit their manuscript again. The perinatal outcome of marginalized mothers and mothers in developed countries cannot be the same. If it is the case, please provide evidence.

---

## [Author Response · Author response to Decision Letter 3]

27 Feb 2025

Date: February 27, 2025.

Dear Editor-in-chief,

Subject: Submission of the corrected systematic review and meta-analysis protocol manuscript.

We wish to extend our gratitude to the editor and reviewers for their valuable and constructive comments and suggestions. Please find attached the author’s point-by-point response letter addressing the editor’s and reviewers’ comments and suggestions. Also attached are a clean protocol manuscript and the tracked changes document for the systematic review and meta-analysis protocol entitled “Feasibility, Acceptability, and Effectiveness of Group Antenatal Care on the Continuum of Care and Perinatal Outcomes in Sub-Saharan Africa: A Systematic Review and Meta-analysis Protocol” with manuscript ID PONE-D-24-40365 by Mengesha MB, et al., for your consideration.

We thoroughly revised the systematic review and meta-analysis protocol in response to the editor’s and reviewers’ comments and suggestions. We confidently assert that the revised protocol meets the publication standards of your esteemed journal. We look forward to your response, appreciate your consideration, and await the outcome of your assessment.

Yours sincerely,

Meresa

Meresa Berwo Mengesha (BSc, MSc, Assistant Professor), Corresponding Author

Department of Midwifery, College of Medicine and Health Sciences,

Adigrat University, Adigrat, Ethiopia.

Email: meresaberu@gmail.com.

Response to reviewers and editor’s comments

Journal Requirements:

Author’s response: Thank you for your critical observation and for bringing this to our attention. We have cross-checked the references twice and confirmed that we have not cited any retracted papers. However, we have made revisions to address some instances of duplication and ensured that the reference is now complete and accurate.

Additional Editor Comments:

Line 64 - 66: "This model is also shown to benefit marginalized women; whose perinatal outcomes are comparable to those in developed countries". While your descriptive response was good, the actual revision you made is still confusing. Please revise this statement.

Author’s response: Dear editor, thank you for your feedback and seeking clarification on this issue. We have revised it accordingly. The statement provided below has also been added to the protocol with track changes. “This model of care also benefits women from marginalized populations who experience maternal and perinatal outcomes comparable to those observed in certain low- and middle-income countries” Please take a look at the revised protocol with the tracked changes.

---

## [Editor Report · Decision Letter 3]

3 Mar 2025

Feasibility, Acceptability, and Effectiveness of Group Antenatal Care on the Continuum of Care and Perinatal Outcomes in Sub-Saharan Africa: A Systematic Review and Meta-analysis Protocol

PONE-D-24-40365R3

Dear Mr Meresa Berwo Mengesha 

We’re pleased to inform you that your manuscript has been judged scientifically suitable for publication and will be formally accepted for publication once it meets all outstanding technical requirements.

Kind regards,

Hale Teka

Academic Editor

PLOS ONE
---

## [Editor Report · Acceptance letter]

PONE-D-24-40365R3

PLOS ONE

Dear Dr. Mengesha,

I'm pleased to inform you that your manuscript has been deemed suitable for publication in PLOS ONE. Congratulations! Your manuscript is now being handed over to our production team.

Kind regards,

on behalf of

Dr. Hale Teka

Academic Editor

PLOS ONE